# High-throughput functional evaluation of *BRCA2* variants of unknown significance

Masachika Ikegami[1,2], Shinji Kohsaka[1✉], Toshihide Ueno [1], Yukihide Momozawa[3], Satoshi Inoue[1,4], Kenji Tamura[5], Akihiko Shimomura[5], Noriko Hosoya [4,6], Hiroshi Kobayashi[2], Sakae Tanaka[2] & Hiroyuki Mano [1✉]

Numerous nontruncating missense variants of the *BRCA2* gene have been identified, but there is a lack of convincing evidence, such as familial data, demonstrating their clinical relevance and they thus remain unactionable. To assess the pathogenicity of variants of unknown significance (VUSs) within *BRCA2*, here we develop a method, the MANO-B method, for high-throughput functional evaluation utilizing *BRCA2*-deficient cells and poly (ADP-ribose) polymerase (PARP) inhibitors. The estimated sensitivity and specificity of this assay compared to those of the International Agency for Research on Cancer classification system is 95% and 95% (95% confidence intervals: 77–100% and 82–99%), respectively. We classify the functional impact of 186 *BRCA2* VUSs with our computational pipeline, resulting in the classification of 126 variants as normal/likely normal, 23 as intermediate, and 37 as abnormal/likely abnormal. We further describe a simplified, on-demand annotation system that could be used as a companion diagnostic for PARP inhibitors in patients with unknown *BRCA2* VUSs.

[1] Division of Cellular Signaling, National Cancer Center Research Institute, Tokyo 104-0045, Japan. [2] Department of Orthopaedic Surgery, Faculty of Medicine, The University of Tokyo, Tokyo 113-0033, Japan. [3] Laboratory for Genotyping Development, RIKEN Center for Integrative Medical Sciences, Kanagawa 230-0045, Japan. [4] Department of Medical Genomics, Graduate School of Medicine, The University of Tokyo, Tokyo 113-0033, Japan. [5] Department of Breast and Medical Oncology, National Cancer Center Hospital, Tokyo 104-0045, Japan. [6] Laboratory of Molecular Radiology, Center for Disease Biology and Integrative Medicine, Graduate School of Medicine, The University of Tokyo, Tokyo 113-0033, Japan. ✉email: skohsaka@ncc.go.jp; hmano@ncc.go.jp

The *BRCA1* and *BRCA2* (*BRCA*) genes encode key proteins in the homology-directed DNA break repair (HDR) pathway, and their inactivation predisposes individuals to cancer development[1]. Germline loss-of-function variants in *BRCA* markedly increase the risk of early-onset breast and ovarian cancer; in such cases, prophylactic oophorectomy and mastectomy and genetic testing for at-risk relatives must be considered[2–4]. Tumors with pathogenic variants within *BRCA* and defective HDR have been shown to be particularly sensitive to platinum-based chemotherapies and poly (ADP-ribose) polymerase (PARP) inhibitors, the efficacy of which is mediated through synthetic lethality in cancer cells with *BRCA* loss-of-function[5,6].

The American College of Medical Genetics and Genomics (ACMG) standards and guidelines for the interpretation of sequence variants recommend a process for variant classification based on criteria using population, computational, functional, and segregation data[7]. The ACMG guidelines assign a categorical strength to each evidence: supporting, moderate, strong, very strong, or stand-alone. Then, each variant is assigned to five categories using combining criteria: benign, likely benign, uncertain significance, likely pathogenic, and pathogenic. Family-based studies including a multifactorial model of pathology, a cosegregation profile, and the cooccurrence and family history of cancer may exemplify the most reliable methods for classifying *BRCA2* gene variants[8,9]. Nonsense or frameshift variants within the coding exons of *BRCA* markedly alter the structures of the protein products and are presumed to confer loss-of-function. However, the vast majority of missense variants are individually rare in both general populations and cancer patients, and case–control studies may not have sufficient statistical significance to classify these variants as pathogenic or benign[10–12]. No current in silico computational prediction algorithm for missense variants is accurate enough when used alone[13]. In the framework of the ACMG guidelines, the functional impact of a variant, which is determined by a well-established functional assay, is regarded as strong evidence for benign/pathogenic status[14]. Thus, the functional evaluation of missense variants of unknown significance (VUSs) is urgently required to improve the interpretation of variants identified by genetic testing and to support clinical decision-making for their carriers[15].

While thousands of *BRCA1* VUSs have been assessed by recently developed high-throughput functional assays[16–18], a few hundred *BRCA2* variants have been evaluated by a conventional functional assay for *BRCA2*, the HDR assay[19,20]. The HDR assay has three issues requiring improvement: (i) the throughput is quite low, (ii) it uses a hamster cell line with transient expression of *BRCA2* cDNA, and most importantly, (iii) it can evaluate only variants in the DNA-binding domain[15,21]. To overcome these limitations, we propose herein a high-throughput method using a human cell line stably expressing *BRCA2* variants that enables the evaluation of all exonic variants of the *BRCA2* gene.

## Results

**Stable transduction of *BRCA2* variants**. The introduction and stable expression of *BRCA2* variants in human cells is technically difficult owing to the relatively large coding sequence of this gene (10.2 kbp)[15]. We addressed this issue by employing a piggyBac transposon vector suitable for the stable introduction of large DNA sequences into the genome[22]. In addition, we utilized a *BRCA2* knockout human colorectal adenocarcinoma cell line, DLD1 *BRCA2* (−/−), which is known to be highly sensitive to PARP inhibition compared to parental cells retaining *BRCA2*[23,24]. If a *BRCA2* missense variant is further introduced into DLD1 *BRCA2* (−/−) cells, the change in PARP inhibitor sensitivity

likely reflects the function of the introduced variant. For instance, the expression of a functionally normal variant in DLD1 *BRCA2* (−/−) cells should restore HDR and thus resistance to PARP inhibitors.

Initially, a total of 107 *BRCA2* variants were selected from a curated database, the BRCA Exchange[25] (Supplementary Data 1). Of these, 32, 10, and 65 variants were classified as benign (Class 1/2), pathogenic (Class 4/5), and VUSs (Class 3), respectively, according to the multifactorial five-tier classification system developed by the International Agency for Research on Cancer (IARC)[10,11,19,26,27]. The IARC classification is based on epidemiological data and does not utilize functional evidence. These *BRCA2* variant cDNAs were generated by site-directed mutagenesis and were then subcloned into the piggyBac vector containing unique 10 bp DNA barcode sequences. These individual piggyBac plasmids, together with the hyPBase transposase expression vector, were transfected into DLD1 *BRCA2* (−/−) cells[28].

The transduction efficiency for 20 selected variants was evaluated by real-time quantitative reverse transcription PCR (qRT-PCR) and digital droplet PCR (Supplementary Fig. 1a, b). The mRNA expression levels were within the physiological range of endogenous *BRCA2* with no significant difference among the variants, whereas the copy numbers of the integrated *BRCA2* cDNA were approximately 10. Western blotting demonstrated that the protein expression levels of the 19 BRCA2 variants were generally equal to that of wild-type BRCA2 (Supplementary Fig. 1c).

**Establishment of MANO-B method**. We next performed a cell viability assay to evaluate the drug sensitivity of the *BRCA2* variants. Cells treated with PARP inhibitors (olaparib, niraparib, and rucaparib) or carboplatin (CBDCA) at various concentrations for 6 days were assessed with PrestoBlue reagent (Supplementary Fig. 2). Wild-type *BRCA2* and benign variants showed greater tolerance to these drugs than pathogenic variants. However, the threshold between the benign and pathogenic variants was unclear. It is important to note that wild-type *BRCA2*-induced cells were significantly more sensitive than parental DLD1 cells; therefore, we should establish a sensitive method in which all comparisons could be made with wild-type-induced cells rather than parental cells. We thus modified the mixed-all-nominated-in-one (MANO) method[29,30] that was originally developed to enable high-throughput evaluation of VUSs in transforming genes and invented the MANO-*BRCA* (MANO-B) method for the functional evaluation of *BRCA2* variants (Fig. 1).

In the MANO-B method, DLD1 *BRCA2* (−/−) cells were transfected with the individual *BRCA2* variant cDNAs and the variant-specific barcodes, mixed, and cultured with or without drugs. Genomic DNA was extracted from each cell mixture after the drug treatments, and the barcode sequences were amplified by PCR and deep sequenced. The barcode read count, which was linearly correlated to the number of viable cells harboring the corresponding variant[29], was normalized to that of the vehicle control to evaluate the relative cell viability. Based on these profiles, the drug sensitivity and pathogenicity of each variant were determined (Fig. 1).

**MANO-B method for 107 *BRCA2* variants**. We performed the MANO-B method using the 107 variants with various concentrations of four drugs. The difference in drug sensitivity among the variants was clearly demonstrated at high drug concentrations (Fig. 2), and the optimal concentrations were thus determined to be 2.0, 0.5, 2.0, and 2.0 μM for olaparib, niraparib, rucaparib, and CBDCA, respectively.

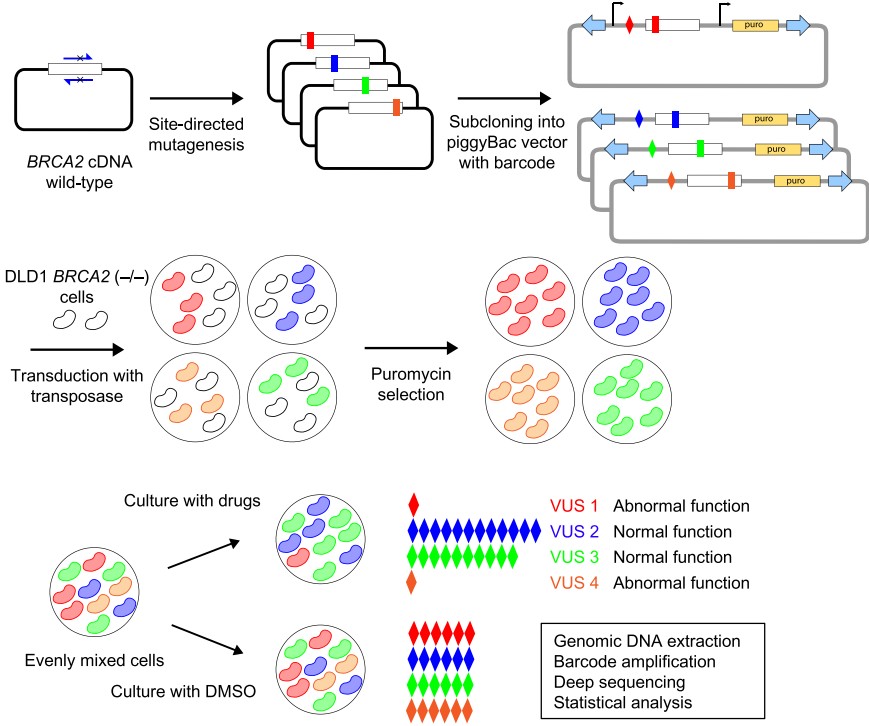

**Fig. 1 Schematic representation of the MANO-B method.** *BRCA2* variants of interest were generated by site-directed mutagenesis and subcloned into the piggyBac transposon vector with a unique barcode sequence, followed by sequence verification. DLD1 *BRCA2* (−/−) cells were individually cotransfected with the *BRCA2* constructs and transposase expression vector. After puromycin selection of stable transfectants, equal numbers of DLD1 cells were mixed and cultured with PARP inhibitors, CBDCA, or DMSO for 12 days. The barcode sequences were amplified by PCR with genomic DNA extracted from the cell mixture and deep sequenced to quantify their relative abundances. The ratio of each barcode abundance was normalized to that of the DMSO-treated control. The relative viability data were analyzed by Bayesian inference, and the pathogenicity of each variant was then classified.

The relative viability of the cells treated with each drug was first calculated by comparing the viability with that of vehicle-treated cells, then normalized to that of cells expressing wild-type *BRCA2*. All variants were sorted by relative viability at the optimal concentrations. A scatter plot of the relative viability of olaparib and the other three drugs revealed an intermediate group including the P2329L, P2639L, D2913H, and S3291C mutants between the benign and pathogenic groups (Supplementary Fig. 3). With this intermediate group as a boundary, all known pathogenic variants showed lower relative viability, whereas all but two known benign variants—R2842H and V2908G—showed higher relative viability. The two variants were indeed shown to be hypomorphic by HDR assay in a previous study[19]. In addition, we found that one likely pathogenic variant (N3187K) defined by ClinVar exhibited normal function as evaluated by the MANO-B method. This discordance between the functional classification and the clinical classification suggests potential errors in the clinical annotations for rare variants.

We believe that the discrepancy in the classification of IARC Class 1/2 variants (R2842H and V2908G) that were evaluated as hypomorphic using the MANO-B method was because of the unreliable evidence which the IARC criteria was based on. The IARC classification is based on epidemiological evidence such as cosegregation data and other family-based genetic analyses[10]. According to the Genome Aggregation Database, both R2842H and V2908G (minor allele frequency = $6.34 \times 10^{-5}$ and $1.69 \times 10^{-5}$) were rare variants. Therefore, it might be difficult to obtain solid evidence for these variants.

ClinVar, a clinical database curated by experts, also classified R2842H and V2908G as benign/likely benign, mainly based on co-occurrence analysis and functional data. R2842H was reported to co-occur with Q3066X, a truncating variant in BRCA2. According to the ACMG standards and guidelines, evidence that a variant is observed in trans with a pathogenic variant for a fully penetrant dominant gene/disorder is a supporting evidence indicating that the variant is benign[7]. However, because a pathogenic BRCA2 variant is not fully penetrant, we believe that co-occurrence with a pathogenic variant cannot exclude the possibility that the variant is hypomorphic. For example, another hypomorphic variant R2784Q was reported to co-occur with pathogenic variants in BRCA1 and BRCA2[26].

Multiple and diverse functional assays for V2908G were reported by Wu et al. in 2005 and Farrugia et al. in 2008, who both belonged to the same research group[31,32]. However, ClinVar does not include two important studies reported by the same group in 2013 and 2018[19,33]. The HDR activity of V2908G in their recent papers was decreased, indicating hypomorph of the variant. No other laboratories have replicated their experiments performed in 2005 and 2008. Thus, we considered V2908G a hypomorphic variant based on their latest reports and our result.

To assess the pathogenicity of such variants accurately, the classification method must be based on a wide variety of evidence. Therefore, we reinterpreted all the 59 Class 1/2/4/5 variants (37 benign and 22 pathogenic) used in the MANO-B method according to the ACMG 2015 guidelines as previously described[27,34]. As a result, 18 variants (31%), including R2842H and V2908G, were classified into VUSs because of insufficient evidence (Supplementary Data 2). Furthermore, N3187K, defined as a likely pathogenic variant by ClinVar, is also defined as VUS according to both IARC and ACMG criteria. The criteria of the ACMG guidelines are more conservative and stringent than those of other classification methods because ACMG uses multiple

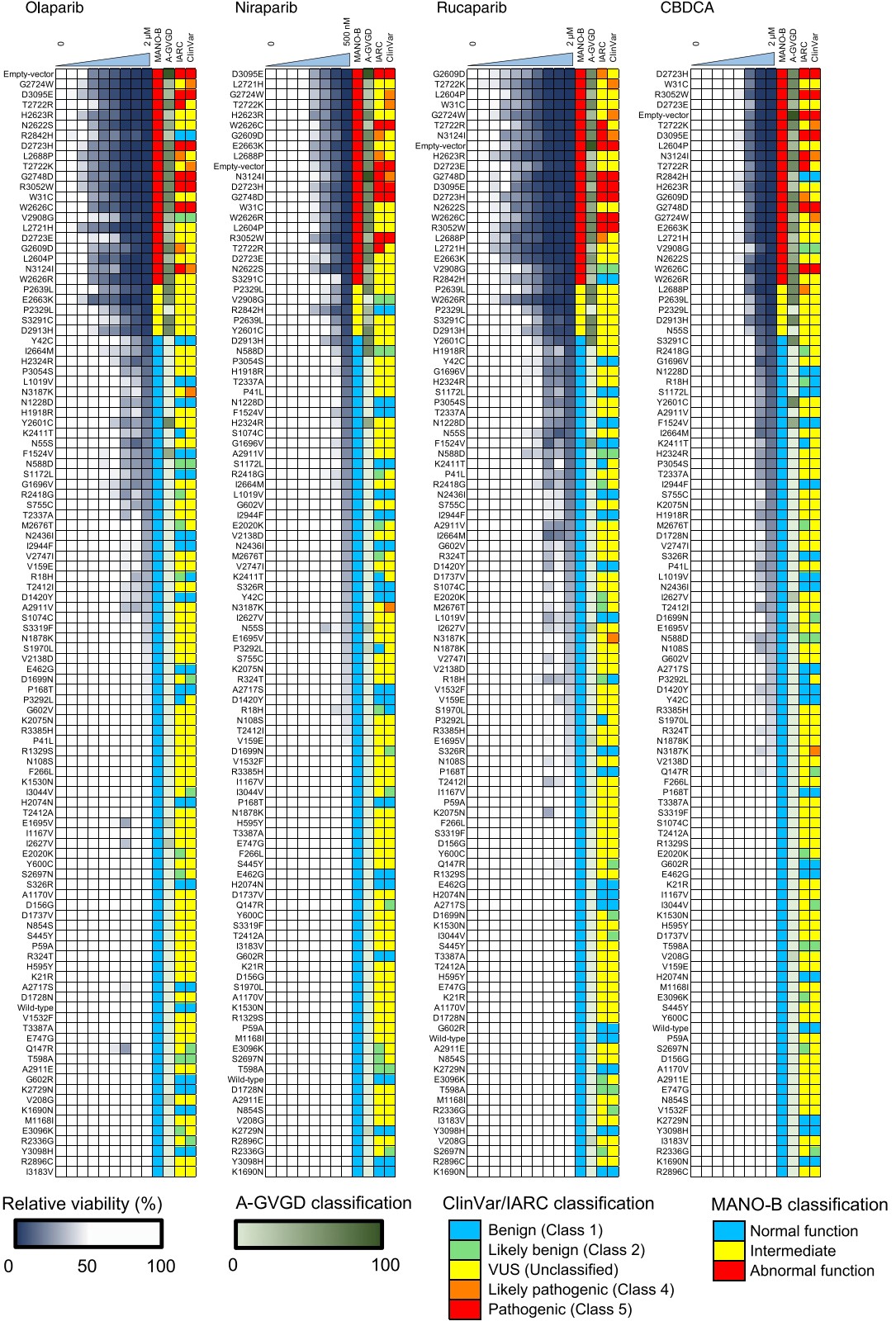

**Fig. 2 Comparative results of the MANO-B method with the IARC, ClinVar, and Align-GVGD classifications for 107 variants.** DLD1 *BRCA2* (−/−) cells transfected separately with the 107 *BRCA2* variants or the empty vector were treated with PARP inhibitors (olaparib, niraparib, and rucaparib), CBDCA, or DMSO at the indicated concentrations. The relative viability of the cells treated with each drug was calculated based on the viability of vehicle-treated cells, then normalized to that of cells expressing wild-type *BRCA2*. All variants were ordered by relative viability at the optimal concentration. The classification of the MANO-B method was based on the relative viability at the optimal concentration of each drug and defined as normal (relative viability of 1–82), intermediate (83–87), and abnormal (88–108). The IARC, ClinVar, and Align-GVGD classifications are color-coded in each column. The ClinVar classifications of five variants (V159E, M1168I, K1530N, G1696V, and A2911E) without assertion criteria were considered as VUSs.

criteria from different points of view and perspectives[7]. This discrepancy among clinical databases implies potential errors in the interpretation of clinical variants. Moreover, the IARC and ACMG classification methods were designed for detecting high-risk pathogenic variants, and therefore, hypomorphic function variants or moderate cancer risk variants could be classified inappropriately[26]. Hence, we regard the discrepancy between functional and epidemiological evidence as reasonable.

To evaluate the evenness of the pooled variant ratio, we collected cell pellets at day 0 (the day the drug treatment was started) to count the barcode reads by MiSeq. Regarding the raw data obtained from batch #1, the minimum, maximum, and median read counts were 778, 13,294, and 5763 for N55S, R3385H, and Y3098H, respectively. The distribution of each variant ratio is shown as a histogram (Supplementary Fig. 4a). A total of 97 variants (90%) were within the range of 0.5–2-fold average. We also calculated the fold change from day 0 to day 12 with dimethyl sulfoxide (DMSO) treatment (Supplementary Fig. 4b). The majority of cells expressing these variants generally exhibited a uniform growth, and the functions of the induced BRCA2 variants did not have much impact on cell growth. The unevenness of the variant ratio probably has a small effect on the result of the MANO-B method because this method is based on fold-change calculation and the abundance bias is corrected through analysis.

**Evaluation of 244 *BRCA2* variants by Bayesian inference.** We expanded the experiment to encompass 244 variants, including 186 VUSs, at the optimal concentration of each drug, and obtained an assay dataset comprising 7344 individual viability values (Supplementary Data 3 and 4). Using the optimal threshold for receiver operating characteristic (ROC) analysis, the estimated sensitivity and specificity of the MANO-B method for variant pathogenicity compared with those of the IARC classification system were 100% and 95% (95% confidence intervals: 85–100% and 82–99%), respectively (Supplementary Fig. 5).

Given that the existence of hypomorphic variants was suggested, a nondichotomous mathematical approach was explored to precisely determine the pathogenicity of each variant. Relative viability data were log normalized and compensated using the values of the wild-type and the D2723H variant as standard values for benign and pathogenic variants, respectively. The resulting viability data followed a two-component Gaussian mixture model by the expectation–maximization algorithm (Supplementary Fig. 6).

In the following analysis, we adopted the Bayes factor (BF) as the strength of evidence in favor of pathogenicity. To calculate the BF, we utilized a Bayesian hierarchical two-component Gaussian mixture model based on the VarCall model[19,35,36] with a noninformative prior probability. In this setting, the BFs were calculated as the probability ratio of a variant being functionally abnormal to it being normal. A five-tiered functional classification based on the BF was assigned to each variant according to established criteria consistent with the ACMG variant evaluation guidelines as follows[37]: fClass 1 (normal; BF ≤ 0.003), fClass 2 (likely normal; 0.003 < BF ≤ 0.053), fClass 3 (intermediate; 0.053 < BF < 18.7), fClass 4 (likely abnormal; 18.7 ≤ BF < 350), or fClass 5 (abnormal; 350 ≤ BF).

The function of a variant, $v$, was estimated by a variant-specific effect, $\eta_v$. Using 22 known pathogenic and 37 known benign variants as the training set, the $\eta_v$ values of benign and pathogenic variants were gradually distributed (Supplementary Fig. 7). The sensitivity and specificity of the Bayesian inference for detecting IARC class 4/5 variants were 95% and 95% (95% confidence intervals: 77–100% and 82–99%), respectively.

When a variant is classified as fClass 4/5, the odds ratio of pathogenicity come out a 17.7:1 in favor of pathogenicity. We also analyzed the concordance between the ACMG classification and the MANO-B classification. All the 41 benign/pathogenic variants (22 benign and 19 pathogenic) used in the MANO-B method were clearly divided into two groups, and the odds ratio of pathogenicity is infinity (Supplementary Fig. 8). Thus, the BF-based functional classification by the MANO-B method provides strong evidence of pathogenicity in the framework of the ACMG guidelines[14,37].

To validate the BF-based classification from the functional perspective, we set two functional classification thresholds determined by the ACMG benign/pathogenic variants to classify the following three components according to a previous report: functionally normal, intermediate, and abnormal[16,19]. The higher threshold between normal and intermediate components is defined as the minimal effect of benign variants according to the ACMG criteria (R18H, $\eta_v = -1.38$), whereas the lower threshold between intermediate and abnormal components is defined as the maximal effect of pathogenic variants (C2535X truncate variant, $\eta_v = -2.17$). As a result, all the functional effects of fClass 1/2 variants were above the higher threshold, whereas all those of fClass 4/5 variants were below the lower threshold. Now, fClass 4/5 variants can be regarded as pathogenic because their probability is in the abnormal component and their functional levels are equivalent to those of the known pathogenic variants.

Notably, the fClass values of the individual variants for the four drugs were concordant (Fig. 3a–c). The data for 0.5 μM niraparib exhibited the highest resolution in the evaluation of the variants of intermediate function. We assume that the difference in functional estimation among drugs in the MANO-B method was primarily because the drug concentrations of olaparib and rucaparib were not as optimal as that of niraparib. The concentration of 2.0 μM of olaparib, rucaparib, and CBDCA was probably too high for DLD1 cells expressing hypomorphic variants to survive, considering that the cell viability assay with 1 μM of olaparib, rucaparib, and carboplatin (Supplementary Fig. 2b) demonstrated broader transitional zones between benign and pathogenic variants. Among the 186 VUSs analyzed in the full dataset with niraparib, 98, 28, 23, 6, and 31 variants were classified as fClass 1–5, respectively (Fig. 3d, Supplementary Data 5). All variants outside functional domains belonged to fClass 1/2. Most abnormal variants were located in the DNA-binding domain (DBD). Other intermediate or abnormal variants were W31C/G/L in the transactivation domain (TAD)[38], P2329L in the MEILB2-binding domain (MBD)[39], and S3291C in the C-terminal domain (CTD), which is a key phosphorylation site for BRCA2–RAD51 interaction[23].

This Bayesian model allowed a combinational classification based on log-normalized relative viability data with prior probability, such as the Align-GVGD classification based on the evolutionary conservation and biophysical properties of amino acids[40] (Supplementary Fig. 9). However, this combinational classification for variants outside the DBD should be interpreted with caution because no reliable prior probability data have been established for this region.

**Model validation.** The internal validation of our model was confirmed by posterior predictive checks. The posterior predictive distribution of the log-normalized relative viability matched the observed data (Supplementary Fig. 10a). Quantile–Quantile (Q–Q) plots showed that the posterior expected standardized residuals of relative viability exhibited a normal distribution (Supplementary Fig. 10b). Furthermore, a strong correlation between the MANO-B method and the HDR assay for 24 selected

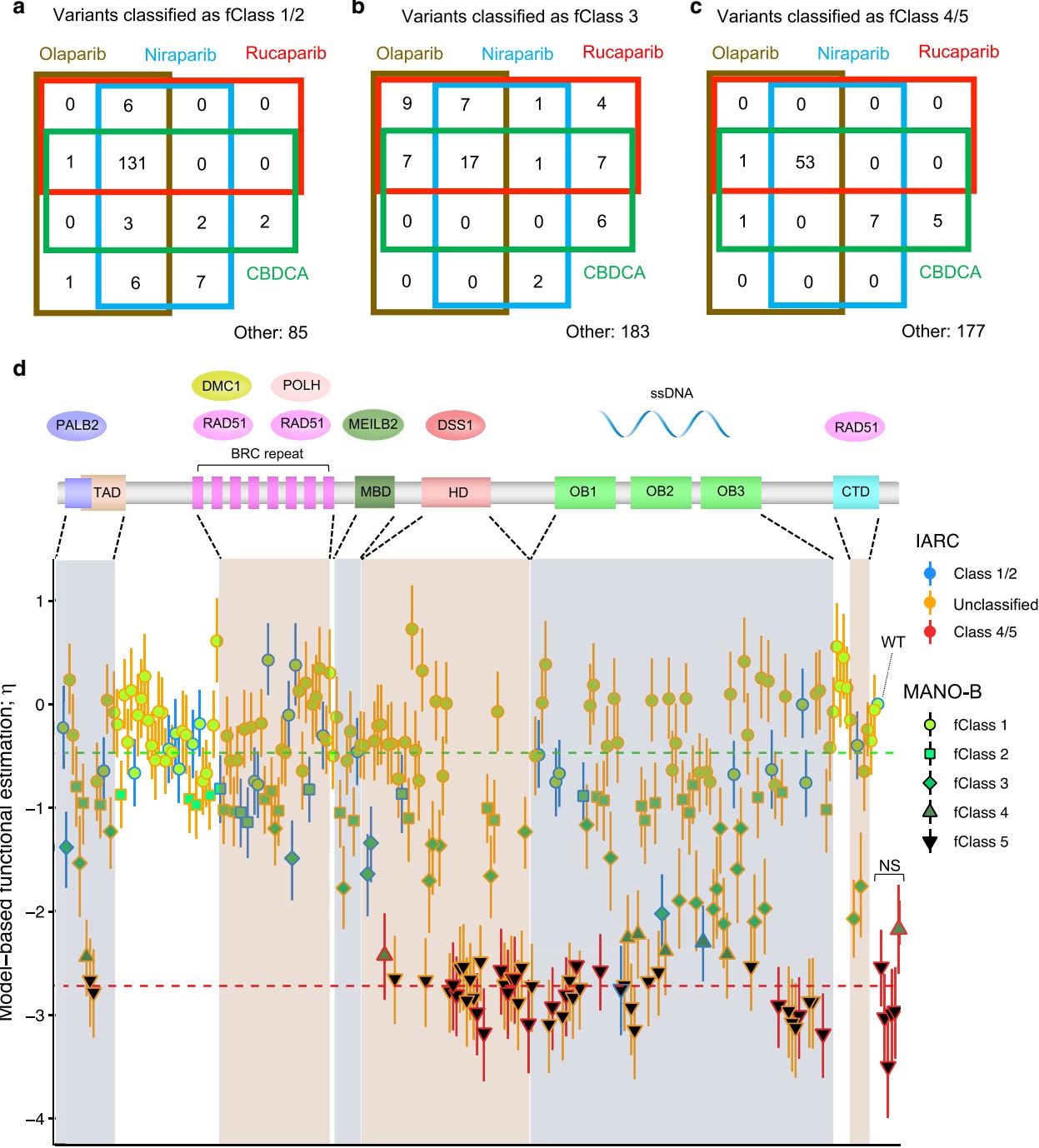

**Fig. 3 Evaluation of 244 *BRCA2* variants by applying Bayesian inference to the MANO-B method. a–c** Functional classification by the MANO-B method with four drugs. A total of 244 *BRCA2* variants were classified by Bayesian inference. The number of variants classified for each drug as fClass 1/2 (**a**), fClass 3 (**b**), and fClass 4/5 (**c**) is shown. No variant was classified as fClass 1/2 with one drug and fClass 4/5 with another drug. **d** Bar plots of functional variant effects and functional diagnosis of 244 *BRCA2* variants mapped against the *BRCA2* full-length sequence and domains. The key domains of *BRCA2* consist of a PALB2 interaction domain encompassing amino acids (a.a.) 10–40, a transactivation domain (TAD) encompassing a.a. 18–105, a RAD51-binding domain including eight BRC repeats encompassing a.a. 1008–2082, an MEILB2-binding domain (MBD) encompassing a.a. 2117–2339, a DNA-binding domain encompassing a.a. 2402–3186 and containing a helical domain (HD) encompassing a.a. 2402–2669, oligonucleotide/oligosaccharide-binding domains (OBs) (OB) encompassing a.a. 2670–2803, 2809–3048, and 3056–3102, and a C-terminal RAD51-binding domain (CTD) encompassing a.a. 3270–3305. The function of each variant was estimated as a variant-specific effect value, $\eta_v$, based on a two-component model. Data of niraparib are shown as a representative case. The upper and lower components correspond to normal function and loss-of-function, respectively. The dotted lines indicate the median value of each component. The classification of each variant based on the Bayes factor (BF) is indicated by the color and shape of the lines and plots, as shown in the legend. Variant functions were classified into five classes by the MANO-B method: fClass 1 (normal) (BF ≤ 0.003), fClass 2 (likely normal) (0.003 < BF ≤ 0.053), fClass 3 (intermediate) (0.053 < BF < 18.7), fClass 4 (likely abnormal) (18.7 ≤ BF < 350), and fClass 5 (abnormal) (350 ≤ BF). $N = 3$ cells were examined over 2 or 3 independent experimental batches. Plots indicate mean values. Open error bars, 95% CIs. WT, wild-type; NS, nonsense variants.

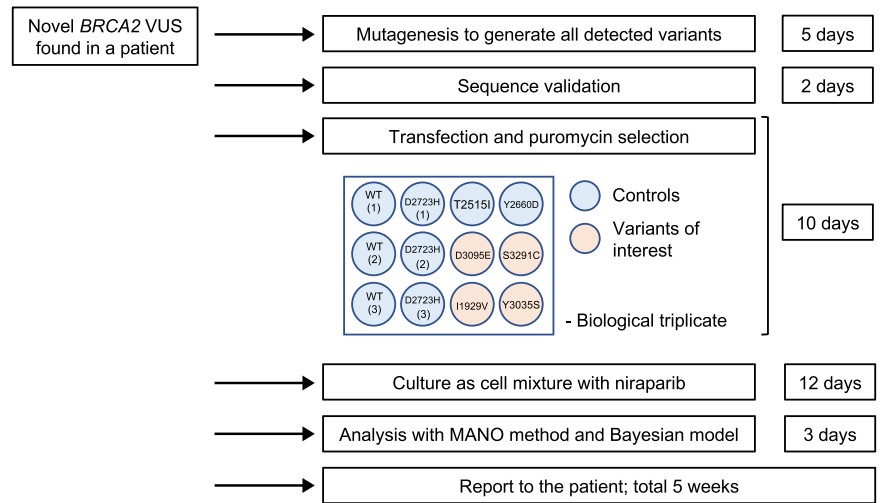

**Fig. 4 The Accurate BRCA Companion Diagnostic (ABCD) test for the clinical functional annotation of novel *BRCA2* VUSs.** Schematic representation of the ABCD test. *BRCA2* variants were investigated by genomic DNA sequencing. The effect of variants on mRNA splicing was evaluated by RNA sequencing to detect the aberrant *BRCA2* transcripts. All detected cDNAs, including four control variants for functionally normal (wild-type and T2515I) and abnormal (Y2660D and D2723H) variants and four variants of interest, were generated for assessment by the small-scale MANO-B method in a 12-well plate. Technical and biological triplicate experiments using niraparib at 0.5 μM are recommended. Data analysis was performed with data from previous experiments, enabling batch effect adjustment. The experimental steps and required times are indicated.

variants was confirmed (Supplementary Fig. 11). The results of the two assays revealed a small discordance in some variants. The S3291C variant exhibited a relatively high value in the HDR assay compared to that in the MANO-B method, probably because S3291 is the phosphorylation site for CDK1/2 and the HDR assay cannot evaluate its function exactly[41].

**Accurate BRCA Companion Diagnostic (ABCD) test.** To rapidly evaluate the pathogenicity of *BRCA2* VUSs in the clinic, the Accurate BRCA Companion Diagnostic (ABCD) test was developed as a potential companion diagnostic for PARP inhibitor treatment. Newly detected *BRCA2* cDNAs, along with four control variants for benign (wild-type and T2515I) and pathogenic (Y2660D and D2723H) variants, were generated for assessment by a small-scale MANO-B method. The resulting data correlate with the data from previous MANO-B experiments, enabling batch effect adjustment for pathogenicity determination. When a *BRCA2* VUS is believed to be related to splicing aberration, its mRNA should be evaluated by RNA sequencing of patient samples[42]. Any aberrant *BRCA2* transcripts detected should also be subjected to the ABCD test (Fig. 4).

An experiment was designed to confirm the accuracy of the ABCD test, in which I1929V, D3095E, Y3035S, and S3291C were assessed as if they were novel variants. The classification of normal and abnormal variants was almost reproducible and consistent with that observed for previous large-scale batches; however, the classification of the D3095E pathogenic variant in batch #1 was underdiagnosed (from fClass 5 to fClass 4, Table 1). One misclassification is crucial in case we judge the clinical application based on the result of a single batch. Because transfection efficiency might cause inconsistent results in the ABCD test, we recognized the importance of multiple independent transfections using different batches of mutants to ensure the accuracy of the assay. Therefore, we combined the results from three batches with different transfections and performed a statistical analysis to annotate the pathogenicity. This approach improved the consistency (4/4, 100%) between the ABCD test

and the large-scale MANO-B method, although the validation study remains to be performed in a larger scale.

## Discussion

Several functional assays for *BRCA2* variants have been developed to date based on the various functions of BRCA2 protein. However, it should be noted that loss of a certain function does not directly lead to cancer predisposition. For example, an established R3052W pathogenic variant did not show pathogenicity in the centrosome amplification assay[32], whereas G2353R, once annotated as a benign variant, showed a pathogenic response in the spontaneous homologous recombination assay[43]. The results of each functional assay should be carefully interpreted together with clinical genetic data.

It is remarkable that the distribution of the functional parameter $\eta_v$ for the MANO-B method was continuous from the normal to the abnormal component and not clearly segregated into two components, in accordance with the HDR assay[19], suggesting the existence of variants of intermediate function[44].

For the nondichotomous classification, a high-resolution quantitative assay together with an appropriate inference method are necessary. In the MANO-B method, all cell clones with individual variants are evenly mixed and cultured in one dish. Since each clone grows competitively under the same culture conditions, including pH, temperature, and drug concentration, we can minimize the technical error and bias and evaluate even small differences in BRCA function among the variants. This is an advantage over other methods such as the cell viability assay and the HDR assay, which test variants individually.

Recently, Findlay et al. reported a robust saturation genome editing method by the use of a haploid cell line, HAP1, that depends on the HDR pathway for growth[16]. In contrast to the MANO-B method, their technology may accurately evaluate the function of variants at the splicing sites or the promoters of *BRCA* genes. On the other hand, our method uses a diploid cell line and, therefore, may better reflect HDR function in physiological conditions. The MANO-B method is, further, able to directly

**Table 1 Results of an experimental ABCD test.**

| Variant | IARC classification | ABCD test | $\eta_v$ | Bayes factor | fClass |
|---|---|---|---|---|---|
| I1929V | Class 1 | Previous batches | 0.37 (−0.02 to 0.79) | $1.05 \times 10^{-5}$ | fClass 1 |
| | | New batch-1 | 0.19 (−0.33 to 0.73) | $1.02 \times 10^{-5}$ | fClass 1 |
| | | New batch-2 | −0.23 (−0.73 to 0.28) | $1.63 \times 10^{-4}$ | fClass 1 |
| | | New batch-3 | 0.08 (−0.43 to 0.6) | $3.71 \times 10^{-5}$ | fClass 1 |
| | | Three batches combined | 0.11 (−0.23 to 0.44) | $1.40 \times 10^{-5}$ | fClass 1 |
| Y3035S | Class 2 | Previous batches | −0.63 (−1.01 to −0.25) | $1.11 \times 10^{-3}$ | fClass 1 |
| | | New batch-1 | −0.55 (−1.05 to −0.04) | $1.12 \times 10^{-3}$ | fClass 1 |
| | | New batch-2 | −0.53 (−1.04 to −0.02) | $9.57 \times 10^{-4}$ | fClass 1 |
| | | New batch-3 | −0.36 (−0.87 to 0.16) | $3.01 \times 10^{-4}$ | fClass 1 |
| | | Three batches combined | −0.47 (−0.79 to −0.15) | $2.79 \times 10^{-4}$ | fClass 1 |
| D3095E | Class 5 | Previous batches | −3 (−3.41 to −2.62) | $1.33 \times 10^{4}$ | fClass 5 |
| | | New batch-1 | −2.31 (−2.8 to −1.78) | $4.45 \times 10^{1}$ | fClass 4 |
| | | New batch-2 | −2.63 (−3.12 to −2.15) | $5.75 \times 10^{2}$ | fClass 5 |
| | | New batch-3 | −2.69 (−3.2 to −2.19) | $9.28 \times 10^{2}$ | fClass 5 |
| | | Three batches combined | −2.52 (−2.85 to −2.19) | $7.85 \times 10^{2}$ | fClass 5 |
| S3291C | Unclassified | Previous batches | −2.07 (−2.43 to −1.69) | $1.15 \times 10^{1}$ | fClass 3 |
| | | New batch-1 | −1.06 (−1.7 to −0.53) | $6.15 \times 10^{-2}$ | fClass 3 |
| | | New batch-2 | −2.06 (−2.6 to −1.34) | $7.05 \times 10^{0}$ | fClass 3 |
| | | New batch-3 | −1.35 (−2.12 to −0.74) | $3.17 \times 10^{-1}$ | fClass 3 |
| | | Three batches combined | −1.46 (−1.86 to −1.11) | $3.39 \times 10^{-1}$ | fClass 3 |

Data are presented as means (95% credible interval).

interrogate the sensitivity of each VUS to different PARP inhibitors. The MANO-B method and the saturation genome editing method may thus be complementary to each other.

While clinical gene testing has identified plenty of *BRCA2* variants, most VUSs are rare and thus there is a lack of substantial familial data. Recently, the uncommon (but not rare) variants G2508S (IARC Class 4) and Y3035S (IARC Class 2) were reported in a large case–control study showing indecisive cancer predisposition[19,44]. These two variants demonstrate the clinical utility of the MANO-B method for clinical annotation of VUSs well. G2508S ($\eta_v = -2.42$, fClass 4) showed hypomorphism, while Y3035S ($\eta_v = -0.63$, fClass 1) showed normal function, although the possibility of splicing aberration by the c.9104A>C single nucleotide variation was not excluded. We consider that G2508S is a hypomorphic variant because the familial data is consistent with the functional result, whereas the pathogenicity of Y3035S is indeterminate because of the conflict among the data. The clinical annotation of such uncommon variants could be assessed by a combinational approach, although additional clinical data are needed to identify the optimal thresholds for the MANO-B method. It is not yet known how to deal with patients harboring *BRCA2* hypomorphic variants. However, we believe these patients may be eligible for PARP inhibitor treatment. Assessing the *BRCA* status may be a good option to predict the effectiveness of PARP inhibitors.

Overall, the MANO-B method is a robust and scalable analysis approach for evaluating the function of *BRCA2* variants. As we developed a dataset of 244 *BRCA2* variants, including controls, additional independent analysis can be merged for precise evaluation using batch effect compensation. Importantly, this technique is applicable to the high-throughput functional assessment of other tumor suppressor genes, augmenting the functional data for rare variants and identifying the association of these variants with cancer susceptibility.

Furthermore, the ABCD test may be useful as a companion diagnostic for PARP inhibitor treatment in patients newly identified as *BRCA2* VUS carriers. There are two scenarios in which the result of the MANO-B method could be applied to patients in the framework of the ACMG guidelines; one is for

predicting the effectiveness of PARP inhibitors for patients with cancer harboring *BRCA2* VUSs, and the other is for cancer risk stratification for considering prophylactic surgery to prevent cancer in patients who at the time are healthy. In the former scenario, clinicians could decide on the use of PARP inhibitors for patients with cancer harboring *BRCA2* VUSs to prevent tumor progression. Therefore, an evaluation of *BRCA* function using the ABCD test could be a companion diagnostic method for PARP inhibitor treatment, although the clear threshold of fClass as a marker to predict which variant is actually beneficial for PARP inhibitor treatment in the clinical setting should be determined by clinical trials. In the latter scenario, healthy patients with hereditary *BRCA2* VUSs can wait and observe for several years for the tumor to emerge. After receiving positive genetic test results, approximately half of the women wait more than 12 months before undergoing prophylactic salpingo-oophorectomy[45]. For patients with fClass 5 variant evaluated by MANO-B method, the application of prophylactic surgery could be considered carefully along with genetic counseling because the possibility of these patients to develop cancer may be high. The penetration rate of cancer likely depends on the extent of *BRCA* function deficiency; healthy patients with a *BRCA* variant of fClass 3 or 4 might be referred to careful check-up rather than prophylactic surgery.

Tumor response to PARP inhibitor therapy has several biomarkers other than *BRCA*1/2 inactivation, such as *EMSY* amplification, Fanconi anemia pathway inactivation, and other HR gene defects[46]. Therefore, it is a good method to evaluate *BRCA* status by whole-genome sequencing of tumors based on mutation and rearrangement signatures without the knowledge of the precise causative mutations[47,48]. Nones et al. demonstrated that 21 breast cancers from 22 *BRCA2* pathogenic variant carriers (95%) in their study exhibited reduced *BRCA* function, whereas only one was *BRCA*-proficient[48]. In contrast to the test of BRCA function, the ABCD test is a simple functional method for evaluating *BRCA2* variant pathogenicity. Although the ABCD test cannot explain the *BRCA* status of non-*BRCA*1/2 tumors, it is a rapid and precise method that directly evaluates the PARP inhibitor sensitivity of cancers harboring *BRCA2* VUSs. In

addition, an advantage of the ABCD test over *BRCA* functional analysis is that the ABCD test can evaluate whether germline *BRCA* variants found in people without cancer are pathogenic variants predisposing to cancer. Therefore, both the ABCD test and the whole-genome sequencing of tumors could be used in parallel for the assessment of *BRCA* status. As the ABCD test reported in this study was limited to *BRCA* functional analysis, we sought to apply this test to other HR genes in the future.

There are several limitations regarding the use of the MANO-B method for clinical annotation. The MANO-B method evaluates HDR function of *BRCA2* variants via PARP inhibitor sensitivity and estimates their pathogenicity indirectly. This method does not directly consider other *BRCA2* functions, such as the regulation of the G2–M transition or transcriptional elongation[49,50]. Although the high concordance of the MANO-B method with the IARC and ACMG classifications suggest that *BRCA2* pathogenicity can be evaluated primarily by investigating HDR activity, we would like to note that BFs for abnormal do not have a direct correlation with the likelihood ratios for pathogenicity, as the normal or abnormal variants defined by this functional assay are not necessarily benign or pathogenic variants in the clinical setting. In fact, there are discrepancies between the functional interpretation by drug sensitivity and clinical annotation of ClinVar in some variants, such as R2842H and V2908G. Those variants were mostly hypomorphic variants or very rare variants that have not been reviewed by expert panels in ClinVar. Therefore, the accurate cancer risks of these variants need to be evaluated carefully. For these reasons, we expect that the functional classification should be used as evidence of pathogenicity and combined with other evidence in a comprehensive framework of ACMG guidelines. We believe that our functional evaluation by drug sensitivity might adjust the interpretation of cancer risks, especially of those variants, although further validation studies are needed.

There are also several limitations of our cDNA-based assay. This approach cannot rule out potential effects on splicing. Importantly, although there are some computational predictors for possible splicing aberrations, either algorithms could not demonstrate adequate reliability for clinical usage, especially for variants outside of the consensus splice sites[51]. Comprehensive splicing functional assays such as hybrid minigene assay and saturation genome editing technique would reinforce the evidence[16,52,53]. It is also recommended to perform RNA-seq of patient blood and tumor to detect aberrantly spliced mRNA. In addition, the MANO-B method was performed in this study using only one colorectal cancer cell line. There is no evidence of the increased risk of colorectal cancer in *BRCA2* pathogenic mutation carriers[54], and it is unclear whether the result of the MANO-B method is reproducible in other cell lines. To evaluate the variant functions precisely and robustly, replication studies with multiple cell lines with various homologous recombination statuses are desired. Although it is noteworthy that the high copy number of integrated BRCA2 constructs was necessary to obtain induced mRNA expression at levels similar to endogenous expression, the regulation of CMV promoter-driven exogenous expression of BRCA2 would not be physiologically relevant.

It is possible that the MANO-B method may lead to a better understanding of cancer biology and provide a concept of clinical functional annotation to improve cancer diagnosis and treatment.

## Methods

**Cell lines and culture conditions**. Human colorectal adenocarcinoma cell line DLD1 parental cells and homozygous *BRCA2* (−/−) variant cells were purchased from Horizon Discovery, Inc. Cells were cultured in RPMI 1640 medium

containing 10% fetal bovine serum and 2 mM L-glutamine at 37 °C in 5% $CO_2$ and passaged at 70–90% confluency with TrypLE Express (all from Thermo Fisher Scientific).

**Choice of variants**. A total of 239 *BRCA2* missense variants identified in the BRCA Exchange database were selected. Sequence nomenclature was based on Human Genome Variant Society (HGVS) and National Center for Biotechnology Information (NCBI) reference NP_000050.2: p.V2466A. We randomly selected 155 variants from functional domains and an additional 89 variants outside functional domains from throughout the coding sequence. Variants in functional domains were selected considering Align-GVGD in silico prediction. Between 14 and 19 VUSs were randomly selected from each of the five Align-GVGD C15–C55 categories, and 62 variants were randomly selected from the C65 category. The proportion of each category was defined as previously reported[19]. An additional five nonsense variants thought to be definitely pathogenic were selected as controls (Supplementary Data 3). Eight VUSs located in the last 3 exonic bases of the splice donor site and five VUSs located in the first 2 exonic bases of the splice acceptor site were possible candidates for disrupting splice consensus sequences. These VUSs were analyzed using Splicing Prediction in Consensus Element (SPiCE) v2.1.3[51], the most accurate in silico splice site prediction algorithm for *BRCA1/2*. Only six variants—V159M (c.475G>A), D23Y (c.67G>T), V211I (c.631G>A), R2336P (c.7007G>C), and R2602T (c.7805G>C)—were predicted to alter canonical splicing, and we assessed the effect of these variants on protein function by the MANO-B method (Supplementary Data 3).

**Plasmid construction**. The piggyBac transposon vector was constructed by inserting a random 10-bp DNA barcode sequence between the CMV promoter and the multiple cloning site of the piggyBac dual promoter vector (PB513B-1; System Biosciences). The *GFP* sequence in the piggyBac vector was deleted by a site-directed mutagenesis technique for the direct repeat-green fluorescent protein (DR-GFP) assay. The full-length wild-type cDNA of human *BRCA2* was subcloned from pcDNA3 236HSC WT (Addgene plasmid # 16246) into the piggyBac vector. Plasmids encoding *BRCA2* variants were developed by a site-directed mutagenesis technique with variant-specific primers designed using an online tool (QuikChange Primer Design; https://www.agilent.com/store/primerDesignProgram.jsp). The primers used for mutagenesis are listed in Supplementary Data 3. For western blot analysis, the N-terminal FLAG-tag (DYKDDDDK) was inserted in the piggyBac-BRCA2 wild-type by site-directed mutagenesis, and then the 19 *BRCA2* variants were also subcloned into the FLAG-BRCA2-piggyBac vector. Plasmids were fragmented by an E220 Focused-ultrasonicator (Covaris, Inc.) to an average size of 300 bp, ligated with adaptors, and sequenced using MiSeq (Illumina). All the entire *BRCA2* cDNA sequences, including the ones with point mutations generated by site-directed mutagenesis, the unique barcode sequences, and the piggyBac vector backbone sequences of every plasmid were confirmed by our original pipeline. Next-generation sequencing (NGS) data were aligned by Bowtie2 v2.3.4.2 against the reference sequence composed of wild-type *BRCA2* cDNA and the piggyBac vector and analyzed with samtools v1.9, bcftools v1.9, and IGV v2.4.10 software (Broad Institute). Plasmids harboring undesirable mutations were discarded. Therefore, all plasmids in this study had only target variants and unique barcode sequences without additional mutations. The hyperactive piggyBac transposase expression vector (pCMV-hyPBase) was provided by Trust Sanger Institute. pCBASceI and pHPRT-DRGFP were a gift from Maria Jasin (Addgene plasmid # 26477 and # 26476, respectively). The puromycin antibiotic selection marker for pHPRT-DRGFP was replaced with the Sh_ble zeocin resistance gene. To construct the piggyBac-DRGFP vector, the ends of the XhoI-SacI-digested DRGFP reporter fragment and the SpeI-ApaI-digested piggyBac transposon vector were blunted and ligated to each other.

**Stable transfection with the piggyBac transposon**. The lipid-based method was used to transfect DLD1 cells with the plasmid constructs. Transfections were performed with exactly one unique plasmid per well. This is essential to guarantee that cells were transduced with only one variant. The same clone of each plasmid harboring a variant was used for all independent batches. Twenty-four hours before transfection, cells were seeded in 96-well plates at a density of $1 \times 10^4$ cells/well. Prior to transfection, the culture medium was replaced with 100 µl of fresh medium. The transfection mixture for individual wells comprised 100 ng of the piggyBac transposon vector, 50 ng of pCMV-hyPBase, 0.5 µl of Lipofectamine Stem Transfection Reagent (Thermo Fisher Scientific), and 50 µl of Opti-MEM I Reduced Serum Medium (Thermo Fisher Scientific). The transfection mixture was added to each well after 20 min of incubation at room temperature. Two days after transfection, cells were cultured with antibiotics (3 µg/ml puromycin for piggyBac-BRCA2 and 100 µg/ml zeocin for piggyBac-DRGFP) for 7 days. Puromycin-resistant polyclonal cell populations harboring piggyBac-BRCA2 were used directly for assays, while a genetically homogeneous cloned cell line harboring piggyBac-DRGFP was generated from a single cell.

**Digital droplet PCR**. Digital droplet PCR was performed using a QX100 Droplet Digital PCR system (Bio-Rad Laboratories) with *BRCA2* cDNA primers and probe and *BRCA2* intron primers and probe, both at a final concentration of 900 nM

primers and 250 nM probe. The sequences are shown in Supplementary Table 1. Aqueous droplets with a 20-μl volume containing final concentrations of 1× Droplet Digital PCR Supermix for probes (Bio-Rad Laboratories) and 100 ng of the genomic DNA template were generated. PCR amplification was conducted using a T100 Thermal Cycler under the manufacturer-recommended conditions. After amplification, the digital PCR data were collected and analyzed using a Bio-Rad QX100 droplet reader and QuantaSoft v1.6.6.0320 software. Crosshair gating was used to automatically split the data into four quadrants by the software's normal setting. Approximately 15,000 droplets were analyzed per well. The induced *BRCA2* cDNA copy number was calculated based on the observation that DLD1 cells are a pseudodiploid human cell line and the copy number of the endogenous *BRCA2* gene is considered to be 2. The 95% confidence intervals were estimated by a Poisson distribution model.

**Real-time quantitative RT-PCR**. Freshly recovered DLD1 *BRCA2* (−/−) cells with 20 variants of *BRCA2*, untransduced DLD1 *BRCA2* (−/−) cells, and DLD1 parental cells were pelleted. Total RNA was extracted from these pellets with RNA-Bee reagent following the manufacturer's instructions. DNase I digestion was performed to minimize DNA contamination, and RNA-Bee purification was repeated to inactivate DNase I. The reverse transcription reaction used 1 μg of total RNA from each sample with SuperScript IV VILO reverse transcriptase. The resulting cDNAs were subsequently used for two-step quantitative RT-PCR with Power SYBR Green PCR Master Mix (Thermo Fisher Scientific) on a 7500HT Fast Real-Time PCR System (Thermo Fisher Scientific). The primers specific for *BRCA2* exon 11 and *ACTB* are shown in Supplementary Table 1. The amplified *BRCA2* sequence was present in parental cells but absent from *BRCA2* (−/−) cells; therefore, mRNA detected in the *BRCA2* (−/−) cells was derived from induced cDNA. The relative *BRCA2* expression levels were normalized to the expression levels of the housekeeping gene *ACTB*. Experiments were performed in technical triplicate, including DLD1 parental cells and DLD1 *BRCA2* (−/−) cells transfected with empty vector control. Then, the expression data were further normalized to the corresponding expression levels in DLD1 parental cells. The results are shown as the averages of biological triplicate experiments.

**Western blotting**. Whole-cell lysates from DLD1 *BRCA2* (−/−) cells harboring 20 variants of the N-terminal FLAG-tagged *BRCA2* cDNA and the empty pig-gyBac vector were prepared with sodium dodecyl sulfate (SDS) sample buffer containing 2-mercaptoethanol (Sigma). The cell lysates (70 μg/sample for FLAG-BRCA2 and 10 μg/sample for beta-Actin) were incubated with the sample buffer for 20 min at 50 °C or for 3 min at 95 °C, respectively, and subjected to SDS polyacrylamide gel electrophoresis on 6% (FLAG-BRCA2) or 10% (beta-Actin) polyacrylamide gels. Precision Plus Protein All Blue Standards (Bio-Rad Laboratories) were run as molecular weight markers along with samples. Electrophoresis was performed at 150 V, and the gels were transferred to an Immobilon-P transfer membrane (Millipore) for 3 h at 50 V. The membranes were blocked with 5% nonfat dry milk in 0.05% Tween 20 containing Tris-buffered saline (TBST) at room temperature for 1 h, and then incubated with a 500-fold diluted primary anti-FLAG antibody (F3165; Sigma) or a 2000-fold diluted primary anti-beta-Actin antibody (#4970; Cell Signaling Technology) in TBST at 4 °C for 16 h. Subsequently, the membranes were incubated with 10,000-fold diluted peroxidase-linked secondary antibody (NA931V for FLAG-BRCA2 and NA934V for beta-Actin; GE Healthcare) at room temperature for 4 h. The target proteins were visualized by an enhanced chemiluminescence reagent, SuperSignal West Femto (GE Healthcare).

**Cell viability assay**. DLD1 *BRCA2* (−/−) cells harboring 20 variants of *BRCA2* cDNA and empty vector, untransduced DLD1 *BRCA2* (−/−) cells, and DLD1 parental cells were seeded in 96-well plates at a density of $2.0 \times 10^3$ cells/well with 100 μl of medium/well, and each drug was added at various concentrations: olaparib (50 nM–5 μM, Selleckchem), niraparib (10 nM–1 μM, Selleckchem), rucaparib (50 nM–5 μM, LC Laboratories), and CBDCA (50 nM–5 μM, Selleckchem). DMSO (Nacalai Tesque) was added to a final concentration of 0.01% (volume/volume) to wells without drugs. Ten microliters of PrestoBlue cell viability reagent (Thermo Fisher Scientific) was added to each well 144 h after exposure to these drugs, and fluorescence intensity was measured with a 2030 ARVO X3 microplate reader and PerkinElmer 2030 Software v4.0 (PerkinElmer) (excitation; 530 nm, emission; 590 nm)[55]. Wells without cells were assessed as the negative controls, and survival data were graphically analyzed as a sigmoid curve by GraphPad Prism software v8.02 for Mac (GraphPad Software, Inc.).

**MANO-B method**. The original MANO method is a high-throughput functional assay previously reported by our laboratory[29,30]. Individually established DLD1 *BRCA2* (−/−) cells with stable *BRCA2* expression were mixed equally and cultured competitively. Cells were seeded at a density of $1 \times 10^4$ cells/cm² on day 0. During the 12-day treatment with PARP inhibitors (olaparib, niraparib, and rucaparib), CBDCA or DMSO, cells were passaged once, on day 6. On day 12, genomic DNA was isolated from cell lysates with a QIAamp DNA Mini Kit (Qiagen). Barcode sequences were amplified by PCR using 300 ng of genomic DNA (100,000-genome and 500,000-barcode equivalent, by the weight of 6 pg of genomic DNA per cell

and an average copy number of integrated cDNA of 10 copies per cell) and primers containing index sequences for deep sequencing (Supplementary Table 2). The quantity and quality of the obtained libraries were evaluated using a Qubit 2.0 fluorometer (Thermo Fisher Scientific) and an Agilent 2200 TapeStation system. Libraries were deep sequenced on the MiSeq sequencer and MiSeq Control Software v2.5.0.5 using a Reagent Kit V2 (300 cycles), and the number of each barcode in each variant was counted. Variants were tested in at least two independent batches in technical triplicate. In every experiment, the wild-type and the D2723H variant were included as the benign and pathogenic controls, respectively. In each independent batch, cells were transfected with the plasmids independently on each occasion because transfection efficiency has a large impact on genomic integration and gene expression, which could affect the result of the MANO-B method. A single clone of each plasmid harboring each desired variant was used for all batches. To remove the concern that a plasmid harbored an additional undesired mutation, we had the entire sequence of each plasmid checked (including the vector backbone) rather than using two independent clones. In batch #1, 107 variants and one empty vector were evaluated. In batches #2 and #3, 244 variants and one empty vector were evaluated. The specific variants are shown in Supplementary Data 1 and 3.

**Model assessment**. The expectation–maximization algorithm was used to determine the appropriate Gaussian mixture model for the data from the MANO-B method with the mclust v5.4.5 package for the R language[56]. The model with the highest Bayesian information criterion was adopted.

**Bayesian hierarchical model analysis**. Barcode counts obtained by the MANO-B method were analyzed using a Bayesian hierarchical mixed model for variant function, which was a modified version of the VarCall model, as previously described[36]. The fold change in the barcode count for variants treated with each drug relative to that for variants treated with the DMSO control was calculated. These values were log normalized to the standard values of $\log_{10}(1.0)$ for wild-type and $\log_{10}(0.003)$ for D2723H as anchors. The remaining 242 variants were unlabeled for pathogenicity in this analysis. We hypothesized that the functional variant-specific effect $\eta_v$ followed a 2-component Gaussian mixture distribution. The prior probability of the variants' pathogenicity inside key domains was noninformative or based on the Align-GVGD classification obtained from the HCI Breast Cancer Genes Prior probabilities website (http://priors.hci.utah.edu/PRIORS) and the BRCA Exchange website (https://brcaexchange.org). The Align-GVGD classification (C0–C65) was defined on the basis of evolutionary conservation of the protein sequence from *Strongylocentrotus purpuratus* to *Homo sapiens*. Key functional domains annotated by the HCI Breast Cancer Genes Prior website were as follows: the PALB2 interaction domain (amino acid residues 10–40), DNA-binding domain (2481–3186), and TR2 RAD51-binding domain (3269–3305). The prior probability outside key domains was estimated at 0.02 by the HCI and the BRCA Exchange, independent of the Align-GVGD classification.

The equations and parameters of the model are as follows:

$$\prod_v \Pr(f_v | D_v, X_v, \theta) =$$

$$\prod_{\{v: D_v = A\}} \Pr(f_v | D_v, X_v, \theta) \prod_{\{v: D_v = N\}} \Pr(f_v | D_v = N, X_v, \theta)$$

$$\prod_{\{v: D_v \text{ is unknown}\}} \left[ \pi_{a(v,A)} \Pr(f_v | D_v = A, X_v, \theta) + \pi_{a(v,N)} \Pr(f_v | D_v = N, X_v, \theta) \right]$$

The terms in the equation above are addressed below.

$\Pr(\text{Data} | \text{Parameters})$ : likelihood of data observation with parameters

$v$ : variant index

$f_v$ : measurements of functional experiments for $v$

$D_v : \begin{cases} D_v = A(v = \text{abnormal}) \\ D_v = N(v = \text{normal}) \end{cases}$

$X_v$ : batch and experimental indices for each measurement

$\theta$ : model parameters

$\pi_{a(v,A/N)}$ : prior probability that is abnormal/normal

The true distributions of the data and parameters were estimated by the formula below.

$$\Pr(f_{v^*} | D_{v^*}, X_{v^*}, \theta) = \prod_{\{(v,b,e): v = v^*\}} \Pr(f_v | D_v, b, e, \theta)$$

We established some constraints and weakly informative prior distributions as described below.

$b$: batch index;

$e$: experimental index;

$\beta_b$: batch-specific random intercept effect;

$\tau_b$: batch-specific random slope effect;

$\kappa_1$: center of the distribution $\beta_b$;

$\kappa_2$: center of the distribution $\tau_b$;

$\lambda_1$: standard deviation of the distribution $\beta_b$;

$\lambda_2$: standard deviation of the distribution $\tau_b$;

$\eta_v$: variant-specific random effect;
$\eta_{abn}$: center of the abnormal variants' $\eta_v$ distribution;
$\eta_{nor}$: center of the normal variants' $\eta_v$ distribution;
$\kappa_1$: standard deviation of the abnormal variants' $\eta_v$ distribution;
$\kappa_2$: standard deviation of the normal variants' $\eta_v$ distribution;
$\psi$: residual error

$$\text{Key domains}: \begin{cases} \text{PALB2 interaction domain (amino acid residues } 10-40) \\ \text{DNA} - \text{binding domain } (2481-3186) \\ \text{TR2 RAD51} - \text{binding domain } (3269-3305) \end{cases}$$

The terms in the formulae above are addressed below.

$$
\begin{aligned}
f_v &\sim \text{Normal}(\beta_b + \tau_b\eta_v, \tau_b\psi) \\
\beta_b &\sim \text{Normal}(\kappa_1, \lambda_1) \\
\tau_b &\sim \text{Normal}(\kappa_2, \lambda_2), \tau_b > 0 \\
\eta_v | D_v = A &\sim \text{Normal}(\eta_{abn}, \sigma_1)\{v : D_v = A \wedge v \neq \text{D2723H}\} \\
\eta_v | D_v = N &\sim \text{Normal}(\eta_{nor}, \sigma_2)\{v : D_v = N \wedge v \neq \text{WT}\} \\
\eta_{WT} &= \log_{10}(1.0) \\
\eta_{D2723H} &= \log_{10}(0.003) \\
\eta_{nor} &= \text{estimated value by mclust package with the training data set} \\
\eta_{abn} &\sim \text{Normal}(\eta_{D2723H}, 5)\,(\text{training data set}) \\
\eta_{abn} &= as\ \beta_b + \tau_b\eta_{abn}\ \text{value is the same in the training data set}(\text{full analysis}) \\
\kappa_1 &\sim \text{Normal}(0, 5) \\
\kappa_2 &\sim \text{Normal}(1, 5) \\
\psi, \sigma_1, \sigma_2, \lambda_1, \lambda_2 &\overset{iid}{\sim} \text{HalfNormal}(0, 5) \\
D_v &\sim \text{Bernoulli}\left(\pi_{a(v)}\right)\{v : D_v \text{ is unknown}\} \\
\pi_{a(v)} &\sim Beta(1,1)(\text{noninformative prior probability})
\end{aligned}
$$

$$
\begin{cases}
\pi_{a(v)} \sim Beta(15.00, 3.48)\{v \in \text{C65, inside key domains}\} \\
\pi_{a(v)} \sim Beta(5.38, 2.57)\{v \in \text{C35, C45, C55, inside key domains}\} \\
\pi_{a(v)} \sim Beta(3.76, 9.00)\{v \in \text{C15, C25, inside key domains}\} \\
\pi_{a(v)} \sim Beta(1.43, 73.1)\{v \in \text{C0, inside key domains}\} \\
\pi_{a(v)} \sim Beta(1.64, 120.44)\{v \in \text{outside key domains}\} \\
\pi_{a(v)} \sim Beta(387, 1.07)\{v \in \text{nonsense variants}\} \\
\qquad (\text{Align} - \text{GVGD} - \text{based prior probability})
\end{cases}
$$

In the above model, the fixed value $\eta_{nor}$ was arbitrarily estimated by the mclust v5.4.5 package with the training set data. In the full analysis, $\eta_{abn}$ was fixed such that the value of $\beta_b + \tau_b\eta_{abn}$ was the same as that in the training data set. Normal($\mu,\sigma$) denotes the normal distribution with mean $\mu$ and standard deviation $\sigma$; $X \sim$ HalfNormal(0,5) indicates that $X$ is normally distributed with a mean of 0 and a standard deviation of 5 but is constrained to be nonnegative. The parameters of the Beta distributions were determined such that their 2.5th and 97.5th percentiles fit the upper and lower 95% confidence limits of the Align-GVGD prediction for each class[40]. The ranges of the normal distributions for the parameters' prior distributions were set to adequately exceed the range of values expected for those parameters. The half-Cauchy distribution was thought to be superior to other weakly informative prior distributions for variance component parameters[57], but the Cauchy distribution might be too broad for our model because very large values could not occur. Thus, a half-normal distribution was adopted for variance component parameters.

**Inference**. A Hamiltonian Monte Carlo algorithm was used to perform inference for the above model. The source code written in the R[58] v3.5.3 and Stan[59] v2.19.3 languages is available (https://github.com/MANO-B/Bayes). We ran four chains of samplers, including 1500 warmup iterations followed by 3500 sampling iterations, and every sampling iteration was adopted. The chain passed the Gelman and Rubin's convergence diagnostic, as the $R$-hat values were less than 1.1 for all parameters[60], indicating convergence across the four chains initiated from disparate starting values, and the simulated posterior values were drawn from the true posterior for each parameter. The marginal posterior means and credible intervals (CIs) of the model parameters, the BF and posterior probabilities of the pathogenic categories for each variant were calculated.

**Posterior predictive checks**. To validate the model's correctness, the fitness of estimation to observed data from 244 variants was statistically checked. QQ plots of the posterior expected standardized residuals of the log-normalized relative viability were generated using the car v3.0.6 package for the R language[61]. Simultaneous 95% CIs are plotted as green dotted lines. To confirm the match between the posterior predictive distribution of $f_v$ and the observed data, we randomly drew 512 parameter samples from the trace. Then, we generated 612 random values per variant from a normal distribution specified by the values of $\eta$, $\beta$, $\tau$, and $\psi$ for each sample, accounting for the batch-to-batch ratio. A posterior predictive density curve of the log-normalized relative viability data was computed from 612 generated data sets containing 512 samples each.

**HDR assay**. Twenty-four *BRCA2* variants encoded individually in the piggyBac vector and an I-SceI expression vector, pCBASce, were cotransfected into DLD1 *BRCA2*

(−/−) cells harboring the DRGFP sequence, according to a previous report[19,33]. Twenty-four hours before transfection, cells were seeded in 12-well plates at a density of $1 \times 10^5$ cells/well. Prior to transfection, the culture medium was replaced with 1 ml of fresh medium. The transfection mixture for individual wells was composed of 1 μg of the *BRCA2* expression vector, 500 ng of the pCBASce vector, 4 μl of Lipofectamine Stem Transfection Reagent, and 250 μl of Opti-MEM I Reduced Serum Medium. The transfection mixture was added to each well after 20 min of incubation at room temperature. I-SceI-induced DNA double-strand breaks in the DRGFP sequence were subsequently repaired by homologous recombination, and intact GFP-expressing cells were produced. Four days after transfection, cells were collected, and GFP-positive cells were counted by fluorescence-activated cell sorting (FACS) using a BD FACSCanto II and BD FACSDiva Software v6.1.3 (BD Bioscience). Every cell count was normalized and rescaled relative to a 1:5 ratio of D2723H:wild-type. All variants were analyzed in biological duplicate and technical triplicate.

**Statistical analysis**. All analyses were performed via the R language. The data are the means ± SD, means ± SEM, or means ± 95% CI, as stated in the figure legends. Exact binomial confidence limits were calculated for test sensitivity and specificity. Differences between variants were determined by the two-sided Kruskal–Wallis rank sum test. $p < 0.05$ was considered statistically significant.

**Reporting summary**. Further information on research design is available in the Nature Research Reporting Summary linked to this article.

## Data availability

The source data underlying Figs. 2 and 3d, and Supplementary Figs. 1a–c, 2, 3, 7–9, and 11a are provided as a Source Data file. All the raw data that support the findings of this study have been deposited in https://github.com/MANO-B/Bayes.

## Code availability

All codes that support the findings of this study are provided as Supplementary Software and available from https://github.com/MANO-B/Bayes.

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

## Acknowledgements

The authors thank A. Maruyama for technical assistance and N. Tanabe and T. Yoshida for collecting data. This study was supported by the World-leading Innovative Graduate Study Program for Life Science and Technology, The University of Tokyo, as part of the WISE Program (Doctoral Program for World-leading Innovative & Smart Education), MEXT, Japan; JSPS KAKENHI grants (#19J13207); grants from the Program for Integrated Database of Clinical and Genomic Information under Grant Number JP18kk0205003, the Leading Advanced Projects for Medical Innovation (LEAP) under Grant Number JP18am0001001, and the Practical Research for Innovative Cancer Control under Grant Number JP18ck0106252 from the Japan Agency for Medical Research and Development, AMED; a grant for Endowed Department (Department of Medical Genomics, Graduate School of Medicine, The University of Tokyo) from Eisai Co., Ltd.

## Author contributions

S.K. and H.M. conceived the project and designed the study. M.I., S.K., and H.M. developed the methodology. M.I. and S.I. performed experiments. M.I. and T.U. analyzed and interpreted the data. Y.M., K.T., A.S., and N.H. provided administrative, technical, or material support. H.K. and S.T. provided experimental and analytical support. M.I., S.K., and H.M. wrote and edited the manuscript with feedback from all authors.

## Competing interests

The authors declare no competing interests.
