## [Peer Review File · Nature Communications]

Reviewers' Comments:

Reviewer #3:

Remarks to the Author:

This is a greatly improved study that largely addressed the major concerns raised in the initial review.

However, there are still discrepancies between the authors' interpretation of the deleterious status of variants with that in ClinVar. On page 9 beginning at line 165, the authors describe why they feel that the classifications in ClinVar for R2842H and V2908G are incorrect and these variants do not behave as neutral functional variants. Importantly, the R2842H variant was evaluated by multiple labs using multifactorial evidence (multiple lines). Key data in support of benign/likely benign is that R2842H was reported to co-occur with a truncating variant in BRCA2 p.Gln3066Ter which is strong evidence for benign status. There is functional data for V2908G reported on ClinVar. "Additionally, numerous and diverse functional assays, such as survival assay by MMC (mitomycin-C) sensitivity, homology directed repair, and centriole amplification assays, showed that this variant behaves like wild-type (Wu_2005 and Farrugia_2008). Furthermore, eight clinical diagnostic laboratories/reputable databases have classified this variant as likely benign in ClinVar. Taken together, the evidence for a benign outcome far exceeds all other evidence, thus the variant is classified as likely benign."

The authors should carefully review all of the evidence in ClinVar and BRCAexchange on why these variants were classified as benign/likely benign and rewrite this section. This discrepancy should also be highlighted more in the discussion as a potential limitation of this approach.

Reviewer #5:

Remarks to the Author:

In this paper, Ikegami et al. describe a method, based on DLD1 cells, to perform functional analysis of BRCA2 missense variants. Variant classification is achieved through the analysis of PARP inhibitor sensitivity data using a Bayesian framework based on a previously-developed model (VarCall).

Variants of uncertain clinical significance are a challenge for the implementation of precision medicine, as carriers of these VUS cannot benefit from preventative and treatment options. Therefore, the subject is timely and is likely to interest a wide readership. Overall, the paper is technically sound and the experiments are conducted with rigor.

The authors have done a good job of responding to the previous reviewers' comments, although in some cases it could be improved.

In particular, responses to comment 2 (from reviewer 3) and Major comment 1 (from reviewer 4) seem not to directly address the questions.

In the first case, it is my understanding that what the reviewer wanted was a discussion about the implications of using PARPi sensitivity for risk stratification, not only for treatment decisions. The authors' response veers towards clinical recommendations that seem out of place. The question about the extent to which drug sensitivity can be used to classify variants according to risk (and the reverse; family-based data associating a variant to risk being used to decide treatment) is a hotly debated issue. The present data suggest that there is a strong correlation between risk and drug response but the correlation is not perfect. I believe that would be an appropriate discussion to have.

It seems to me that Major comment 1 (from reviewer 4) was casting doubt on the odds generated by the assay (in line with the reviewers' critique of VarCall in comment 3) and not about the discrepancy between genetic and clinical versus functional data.

I believe these two points should be clarified and they illustrate a concern that the paper is very dense and lacks clarity with references to IARC and ACMG rules that are difficult to follow even for someone familiar with the field.

The problem is primarily that important limitations, most recognized by the authors, may be buried and difficult for the reader to identify. Specifically, the use of a colorectal cancer cell line, the use of drug sensitivity to classify variants according to risk, the high copy number of integrated BRCA2 constructs, etc.

Minor issues:

The authors did improve the terminology but I suggest that, in most instances, 'mutation' should not be used and 'variant' (with a qualifier; 'pathogenic' or 'benign') should be used instead. Also, for functional classifications, the use of 'functional impact' would be preferred to instead of 'deleterious'.

When reporting specificity and sensitivity, please also report the confidence intervals.

With some published papers assessing thousands of variants and several other assessing hundreds at a time, I am not sure this assay and the data reported can be considered high-throughput.

Response to the reviewer's comments

Response to Reviewer #3:

We deeply thank you for the warm encouragement and for the thoughtful comments to improve our paper. Taking your suggestions into consideration, we have revised our paper as described below.

Comment: This is a greatly improved study that largely addressed the major concerns raised in the initial review. However, there are still discrepancies between the authors' interpretation of the deleterious status of variants with that in ClinVar. On page 9 beginning at line 165, the authors describe why they feel that the classifications in ClinVar for R2842H and V2908G are incorrect and these variants do not behave as neutral functional variants. Importantly, the R2842H variant was evaluated by multiple labs using multifactorial evidence (multiple lines). Key data in support of benign/likely benign is that R2842H was reported to co-occur with a truncating variant in BRCA2 p.Gln3066Ter which is strong evidence for benign status. There is functional data for V2908G reported on ClinVar. "Additionally, numerous and diverse functional assays, such as survival assay by MMC (mitomycin-C) sensitivity, homology directed repair, and centriole amplification assays, showed that this variant behaves like wild-type (Wu_2005 and Farrugia_2008). Furthermore, eight clinical diagnostic laboratories/reputable databases have classified this variant as likely benign in ClinVar. Taken together, the evidence for a benign outcome far exceeds all other evidence, thus the variant is classified as likely benign."

The authors should carefully review all of the evidence in ClinVar and BRCAexchange on why these variants were classified as benign/likely benign and rewrite this section. This discrepancy should also be highlighted more in the discussion as a potential limitation of this approach.

Reply: Thank you very much for your kind comments that raise an important point. As you mentioned, R2842H was reported to co-occur with Q3066X, a truncating variant in BRCA2. According to the ACMG standards and guidelines, an evidence that a variant is observed in trans with a pathogenic variant for a fully penetrant dominant gene/disorder is a supporting evidence indicating that the variant is benign. However, because a pathogenic BRCA2 variant is not fully penetrant, we believe that the co-occurrence with a pathogenic variant cannot exclude the possibility that the variant is hypomorphic. For example, another hypomorphic variant R2784Q was reported to co-occur with pathogenic variants in BRCA1 and BRCA2 (Guidugli L et al, 102, 233-248, AJHG, 2018 and Mesman LS et al, 2, 293-302, Genet Med, 2019).

As you also pointed out, multiple and diverse functional assays for V2908G were reported by Wu K et al in 2005 (65, 417-26, *Cancer Res*, 2005) and Farrugia DJ et al in 2008 (68, 3523-31, *Cancer Res*, 2008), who belonged to the same research group. However, ClinVar does not include two important studies reported by the same group (Guidugli L et al, 73, 265-75, *Cancer Res*, 2013 and Guidugli L et al, 102, 233-248, *AJHG*, 2018). In their papers, the HDR activity of V2908G is not consistent. In 2005, Wu K et al reported that the HDR activity, MMC sensitivity, and centrosome amplification activity of V2908G were normal. In 2008, Farrugia reported that the HDR fold activity (WT = 5.0, D2723H pathogenic variant = 1.0) was 6.0 and that the centriole amplification activity of V2908G was normal. However, Guidugli L et al reported that the HDR fold activity of V2908G was 3.94 in 2013 and 2.44 in 2018. They did not mention why this sequential decrease in HDR activity was observed. In addition, no other laboratories have replicated their experiments performed in 2005 and 2008. Thus, we regarded V2908G as a hypomorphic variant based on their latest reports and our result.

Based on a careful review of the evidence, we have added a paragraph regarding the discrepancy between our interpretation of the functional status of variants and the clinical classification in ClinVar in the section entitled “MANO-B method for 107 BRCA2 variants” (page 9, lines 171–187).

We have also added described this discrepancy as a limitation of our method in the Discussion (page 20, line 426–434).

Response to Reviewer #5:

We sincerely appreciate your warm encouragement and great suggestions to strengthen our study. Taking your suggestions into consideration, we have revised our paper as described below.

General comment:

In this paper, Ikegami et al. describe a method, based on DLD1 cells, to perform functional analysis of BRCA2 missense variants. Variant classification is achieved through the analysis of PARP inhibitor sensitivity data using a Bayesian framework based on a previously-developed model (VarCall). Variants of uncertain clinical significance are a challenge for the implementation of precision medicine, as carriers of these VUS cannot benefit from preventative and treatment options. Therefore, the subject is timely and is likely to interest a wide readership. Overall, the paper is technically sound and the experiments are conducted with rigor. The authors have done a good job of responding to the previous reviewers' comments, although in some cases it could be improved. In particular, responses to comment 2 (from reviewer 3) and Major comment 1 (from reviewer 4) seem not to directly address the questions.

Major comment 1: In the first case, it is my understanding that what the reviewer wanted was a discussion about the implications of using PARPi sensitivity for risk stratification, not only for treatment decisions. The authors' response veers towards clinical recommendations that seem out of place. The question about the extent to which drug sensitivity can be used to classify variants according to risk (and the reverse; family-based data associating a variant to risk being used to decide treatment) is a hotly debated issue. The present data suggest that there is a strong correlation between risk and drug response but the correlation is not perfect. I believe that would be an appropriate discussion to have.

Reply: Thank you very much for your suggestion. As you pointed out, there are discrepancies between the functional interpretation by drug sensitivity and the clinical annotation of ClinVar in some variants, such as R2842H and V2908G. Those variants were mostly hypomorphic or very rare variants that have not been reviewed by an expert panel in ClinVar. Therefore, the accurate cancer risks of these variants need to be evaluated carefully. Thus, we expect that the functional classification should be used as evidence of pathogenicity and combined with other evidence in a comprehensive framework of ACMG guidelines. We believe that our functional evaluation by drug sensitivity might adjust the interpretation of cancer risks, especially of those variants, although further validation studies are needed. We have added these sentences in the Discussion (page 20, lines 426–434).

Major comment 2: It seems to me that Major comment 1 (from reviewer 4) was casting doubt on the odds generated by the assay (in line with the reviewers' critique of VarCall in comment 3) and not about the discrepancy between genetic and clinical versus functional data.

Reply: Thank you very much for your comment. The odds ratio of the positive result by the MANO-B method is 17.7:1 (compared with the IARC classification) or infinity (compared with the ACMG classification), in favor of pathogenicity. This can be regarded as strong evidence in the framework of ACMG guidelines. We have added several sentences in the Results section (page 12, lines 247–248, lines 251–254).

Major comment 3: I believe these two points should be clarified and they illustrate a concern that the paper is very dense and lacks clarity with references to IARC and ACMG rules that are difficult to follow even for someone familiar with the field.

Reply: Thank you very much for your comment. To clarify the IARC and ACMG rules, we have added the following three sentences:

The ACMG guidelines assign a categorical strength to each evidence: supporting, moderate, strong, very strong, or stand-alone. Then, each variant is assigned to 1 of the following 5 categories using combining criteria: benign, likely benign, uncertain significance, likely pathogenic, and pathogenic (page 4, lines 65–68).

In the framework of the ACMG guidelines, the functional impact of a variant, which is determined by a well-established functional assay, is regarded as partial evidence for a benign/pathogenic status (page 4, lines 77–79).

The IARC classification is based on epidemiological data and does not utilize functional evidence (page 6, lines 108–109).

Major comment 4: The problem is primarily that important limitations, most recognized by the authors, may be buried and difficult for the reader to identify. Specifically, the use of a colorectal cancer cell line, the use of drug sensitivity to classify variants according to risk, the high copy number of integrated BRCA2 constructs, etc.

Reply: Thank you very much for your advice. We have clarified the indicated potential limitations in the Discussion as follows:

There is no evidence of the increased risk of colorectal cancer in BRCA2 pathogenic mutation carriers, and it remains unclear whether the result of the MANO-B method is reproducible in other cell lines (page 20, line 443–444).

In fact, there are discrepancies between the functional interpretation of some variants, such as R2842H and V2908G, with the clinical annotation in ClinVar. Thus, there is a question regarding the extent to which drug sensitivity can be used to classify variants according to cancer predisposition risk. Although our data suggest that there is a strong correlation between cancer risk and drug response, the correlation is not perfect. Therefore, the functional

classification should be used as partial evidence of pathogenicity and combined with other evidence in a comprehensive framework, such as the ACMG guidelines. In addition, accurate cancer risks of hypomorphic variants are still unknown. Further studies are needed to confirm the functional results with the extent of cancer predisposition risk (page 20, lines 425–433). It is noteworthy that the high copy number of integrated BRCA2 constructs was necessary to obtain induced mRNA expression at levels similar to endogenous expression. The regulation of CMV promoter-driven exogenous expression of BRCA2 would not be physiologically relevant (page 20, lines 447–450).

Minor issues:

Minor comment 1: The authors did improve the terminology but I suggest that, in most instances, ‘mutation’ should not be used and ‘variant’ (with a qualifier; ‘pathogenic’ or ‘benign’) should be used instead. Also, for functional classifications, the use of ‘functional impact’ would be preferred to instead of ‘deleterious’.

Reply: Most of the word ‘mutation’ has now been replaced with ‘variant’ in the updated manuscript. We also reclassified functional impact of variants as “normal,” “intermediate,” or “abnormal”, according to the paper of Brnich SE et al (Genome Medicine, 2020. doi 10.1186/s13073-019-0690-2).

Minor comment 2: When reporting specificity and sensitivity, please also report the confidence intervals.

Reply: Exact binomial confidence limits were calculated for test sensitivity and specificity, and we added them appropriately (page 11, line 223 and page 12, line 247)

Minor comment 3: With some published papers assessing thousands of variants and several other assessing hundreds at a time, I am not sure this assay and the data reported can be considered high-throughput.

Reply: Thank you very much for your suggestion. We agree that the saturation genome editing technique by Findlay GM et al (562, 217-222, Nature, 2018) and multiplex homology-directed DNA repair assay by Starita LM et al (103, 498-508, Am J Hum Genet, 2018) can assess thousands of variants at a time. Their methods are based on an *a priori* screening approach and uses a pooled SNV library for generating variants. In contrast, our method is based on a *post hoc* approach and utilizes non-mixed vectors consisting of individual target variants. Compared with other *post hoc* assays, such as the DR-GFR method, we think that the MANO-B method, which can assess hundreds of variants at a time, is a high-throughput assay.